# High-density volumetric super-resolution microscopy

Sam Daly ®[1], João Ferreira Fernandes ®[2], Ezra Bruggeman ®[1], Anoushka Handa[1], Ruby Peters[3], Sarah Benaissa[4], Boya Zhang[4], Joseph S. Beckwith ®[1], Edward W. Sanders ®[1], Ruth R. Sims ®[5], David Klenerman ®[1], Simon J. Davis[2], Kevin O'Holleran[4] & Steven F. Lee ®[1] ✉

Volumetric super-resolution microscopy typically encodes the 3D position of single-molecule fluorescence into a 2D image by changing the shape of the point spread function (PSF) as a function of depth. However, the resulting large and complex PSF spatial footprints reduce biological throughput and applicability by requiring lower labeling densities to avoid overlapping fluorescent signals. We quantitatively compare the density dependence of single-molecule light field microscopy (SMLFM) to other 3D PSFs (astigmatism, double helix and tetrapod) showing that SMLFM enables an order-of-magnitude speed improvement compared to the double helix PSF by resolving overlapping emitters through parallax. We demonstrate this optical robustness experimentally with high accuracy ( $> 99.2 \pm 0.1\%$, 0.1 locs $\mu m^{-2}$) and sensitivity ( $> 86.6 \pm 0.9\%$, 0.1 locs $\mu m^{-2}$) through whole-cell (scan-free) imaging and tracking of single membrane proteins in live primary B cells. We also exemplify high-density volumetric imaging (0.15 locs $\mu m^{-2}$) in dense cytosolic tubulin datasets.

Single-molecule localization microscopy (SMLM) is a super-resolution technique that separates the fluorescence emission of individual fluorophores temporally to observe biological systems with sub-diffraction resolution[1–4]. Direct imaging in three dimensions (3D) enables the study of complex biological morphologies and dynamic processes that would otherwise be underestimated in 2D.

In SMLM, the fluorescence from a single emitter is observed as a diffraction-limited spot on a detector, known as the *point spread function* (PSF). Generally, 3D-SMLM employs optical elements that transform the standard 2D PSF into spatial distributions that also encode axial position. These 3D PSFs exhibit lateral spatial footprints that are much larger in area than the standard PSF, meaning the projection of a 3D volume onto a 2D detector usually necessitates considerably slower acquisition rates (typically 5 to 10-fold) due to a higher likelihood of PSF overlap[5,6]. However, the number of emitters localized per frame governs imaging speed, and therefore dense emitter datasets are desirable. This is exemplified in recent work from Legant et al. where impressive super-resolved whole-cell volumes were obtained over very long acquisition times (i.e., 3–10 days)[7,8]. This extended experimental duration was necessary to generate an image with a resolution comparable to a corresponding electron microscope experiment[8].

Long imaging durations present unrealistic conditions for typical cellular experiments and also reduce the quantity of biological repeats that can be performed within appropriate timescales. While strategies exist to reduce PSF overlap—such as specialized labeling protocols and post-processing algorithms[9–11]—they are ultimately limited by the decrease in lateral resolution at the expense of a greater depth-of-field

[1]Yusuf Hamied Department of Chemistry, University of Cambridge, Lensfield Road, Cambridge CB2 1EW, UK. [2]Radcliffe Department of Medicine and MRC Human Immunology Unit, John Radcliffe Hospital, University of Oxford, Oxford OX3 9DU, UK. [3]Department of Physiology, Development, and Neuroscience, University of Cambridge, Cambridge CB2 3EL, UK. [4]Cambridge Advanced Imaging Centre, Downing Site, University of Cambridge, Cambridge CB2 3DY, UK. [5]Wavefront-Engineering Microscopy Group, Photonics Department, Institut de la Vision, Sorbonne Université, INSERM, CNRS, Institut de la Vision, Paris, France. ✉e-mail: sl591@cam.ac.uk

(DoF). However, a recent study revealed a lack of post-processing solutions specifically for dense 3D datasets[12]. Hence, to have broad applicability to the biological community there is a fundamental need for robust strategies to perform 3D-SMLM at high densities. This will bring 3D-SMLM into line with the timescales and workflows of current 2D cellular experiments and is another important step toward real-time 3D-SMLM.

Sub-diffraction axial precision can be achieved by engineering the shape of the PSF to simultaneously encode the lateral and axial position of a single emitter in a 2D image[13,14]. A variety of engineered PSFs have been reported, including astigmatism (~1 µm DoF)[15], a bisected pupil (~1 µm DoF)[16], the corkscrew PSF (~3 µm DoF)[17], the double helix PSF (~4 µm DoF)[5,18,19], and the tetrapod PSF (6–20 µm DoF)[20,21]. On the other hand, single-molecule light field microscopy (SMLFM)[22] is an SMLM technique that places a refractive microlens array (MLA) in the back focal plane (BFP) of a widefield microscope to encode 3D position into the PSF[23,24]. SMLFM is wavelength non-specific, possesses a large tuneable DoF, and high photon throughput[13], and the PSF can be fitted with conventional 2D algorithms. The unique advantage of SMLFM is that it operates through parallax whereby the PSF is comprised of several spatiotemporally correlated perspective views displaced in proportion to the curvature of the wavefront. As such, SMLFM is particularly suited to high spot densities for two key reasons: (1) single emitters that occur at different axial planes (but overlap laterally) are imaged at different locations in different perspective views and can be distinguished, and (2) we illustrate a redundancy whereby a localization is not required in every perspective view to be localized in 3D.

Multi-focal plane microscopy also segments the BFP to image two or more focal planes and capture 3D volumes[25–28]. However, this work will focus on techniques that yield single-snapshot sub-diffraction axial precision over extended axial ranges.

Here, we report single-snapshot 3D super-resolution imaging over an 8 µm DoF using a hexagonal MLA. We quantitatively compare the performance of SMLFM to other common 3D PSFs as a function of spot density through simulations. We then apply SMLFM experimentally to the scan-free imaging and tracking of individual B-cell receptors and the imaging of tubulin in HeLa cells to show that overlapping PSFs minimally affect the localization precision and that existing labeling strategies can now be directly transferred to 3D imaging pipelines.

## Results and discussion
### Density dependence of 3D PSFs
Current state-of-the-art 3D PSFs are typically created by phase modulation in the BFP of the objective lens (i.e., with a phase mask). This phase modulation gives rise to spatial distributions of intensity in the imaging plane that change as a function of the axial position of the emitter. These changing PSFs can be understood in the context of high-density imaging by collapsing the entire PSF onto the 2D detector, which we define as the *PSF footprint*, see Fig. 1a(i–iii). Alternatively, in SMLFM the MLA segments the BFP and focuses an array of spots on the detector as shown in Fig. 1b (also see Supplementary Note 1).

Raw localization datasets were simulated for the SMLM modalities presented in Fig. 1 (standard, astigmatic, double helix, light field, and tetrapod) to investigate the effect the of PSF footprint on the ability to resolve single emitters at high densities (Fig. 2a). Briefly, the emitter density ($\rho_{loc}$) of simulated SMLM data was systematically increased from 0.005 µm$^{-2}$ (2 localizations per 20 µm × 20 µm field-of-view, FoV) to 0.375 µm$^{-2}$ (150 localizations) and subsequently processed using

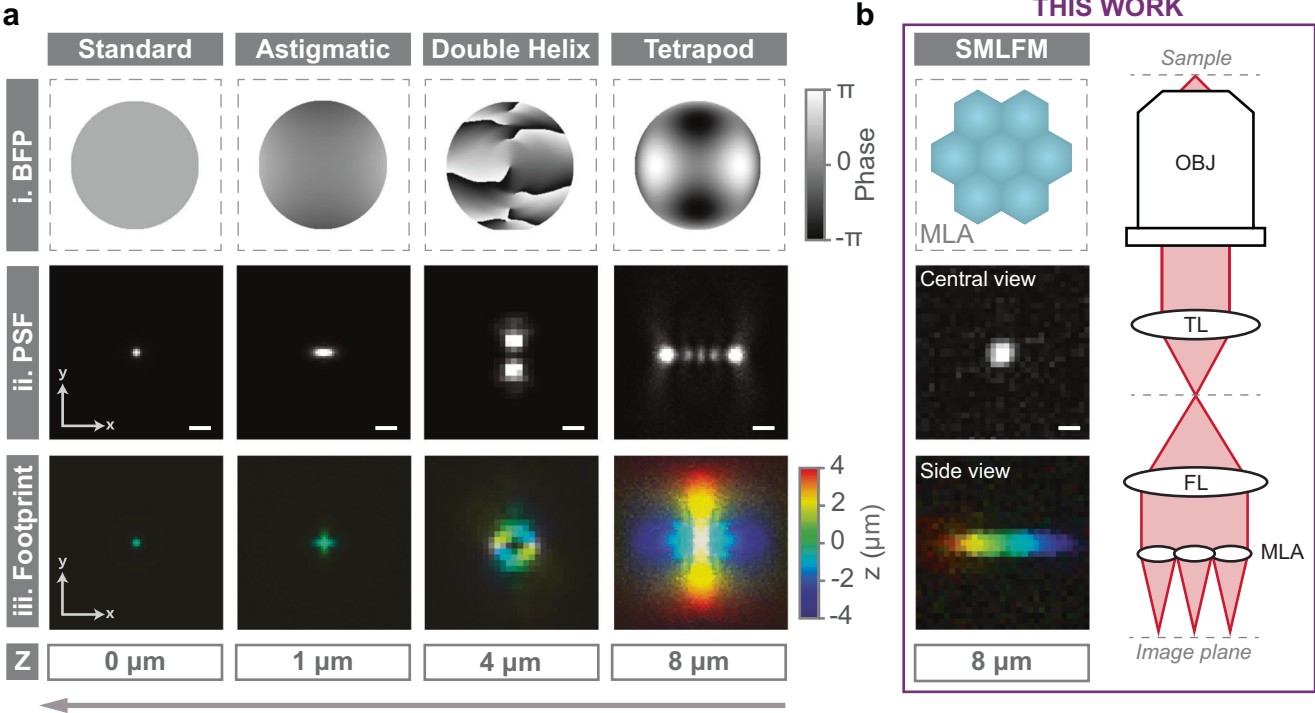

**Fig. 1 | Encoding the 3D position of single-molecule fluorescence into a 2D image. ai** The standard (2D) point spread function (PSF) can be modified to encode 3D position by phase modulation in the back focal plane (BFP, indicated by gray level) of a widefield microscope. **aii** Key 3D SMLM techniques include astigmatism, the double helix PSF, and the tetrapod PSF, shown here in an 8 × 8 µm$^2$ field-of-view (scale bars are 1 µm). **aiii** The associated PSF footprints integrated over their entire axial range (color-coded by depth). The loss in lateral resolution at the expense of axial range leads to a lower signal-to-noise ratio. **b** Schematic of the microlens array used in this SMLFM platform, the PSF in the central perspective view, and the PSF footprint integrated over the entire 8 µm axial range (color-coded by depth). A simplified optical diagram of SMLFM is also shown on the right, where OBJ = objective, TL = tube lens, FL = Fourier lens, and MLA = microlens array. Optical diagrams for all the 3D techniques discussed herein can be found in Supplementary Fig. 2. Pixel size is 110 nm for standard, astigmatism and the tetrapod PSF, and 266 nm for the DHPSF and SMLFM to reflect experimental parameters.

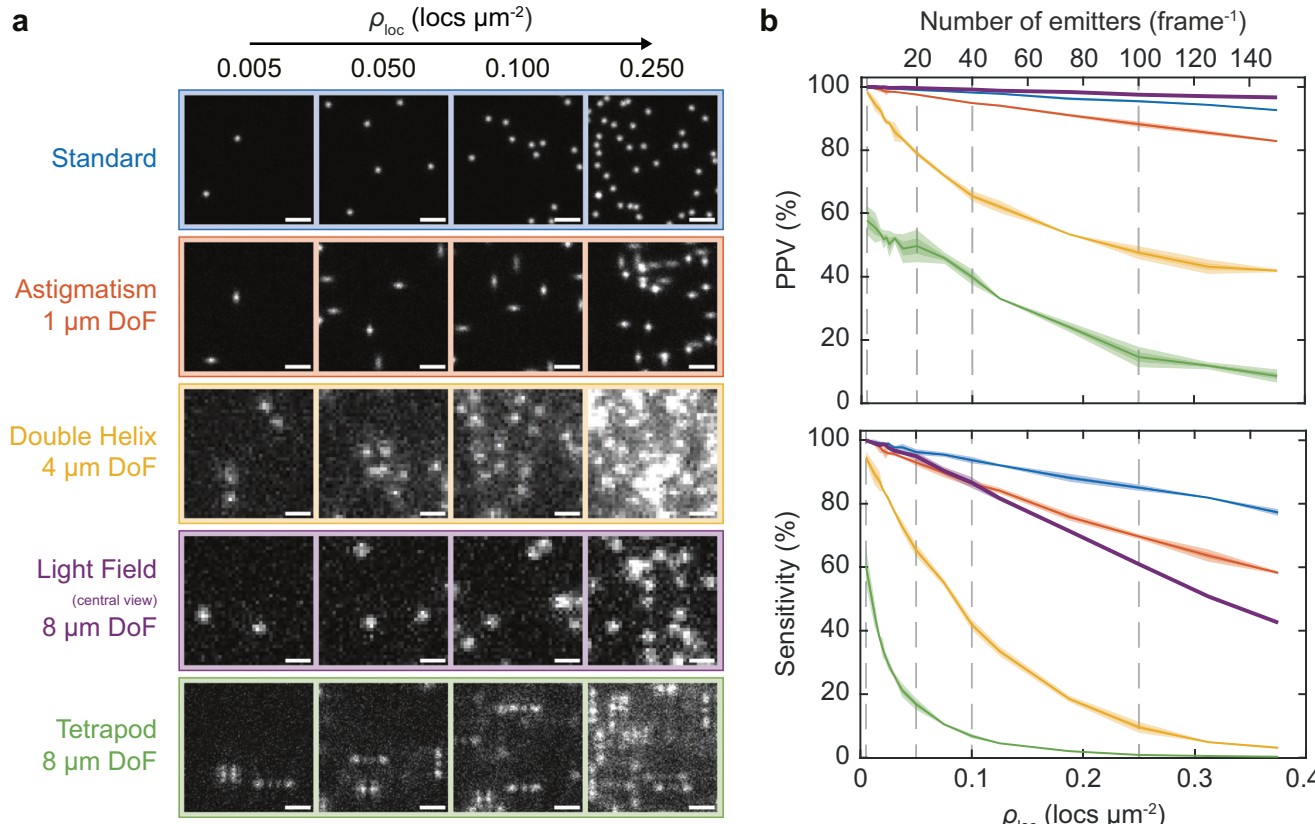

**Fig. 2 | SMLFM consistently outperforms other 3D-SMLM techniques at correctly identifying and reconstructing single emitters at increasing densities.**
**a** Representative snapshots of simulated raw localization data (100 frames, $n = 3$) in a $10 \times 10\ \mu m^2$ zoomed region for each imaging modality discussed herein (2D, astigmatism, double helix PSF, light field [central view] and tetrapod PSF). The scale bar represents $2\ \mu m$. DoF indicates the depth of field achieved by each technique.
**b** Top: Average positive predictive value (PPV) curves for each SMLM technique as a function of emitter density ($\rho_{loc}$) at 4000 detected photons, where PPV refers to the

number of true positive localizations vs. total number of fitted localizations. Bottom: Average sensitivity curves as a function of $\rho_{loc}$ at 4000 detected photons, where sensitivity refers to the number of true positive localizations vs. the total number of ground-truth localizations. Light and dark-shaded regions represent the first and second standard deviations from the mean over three repeats of 100-frame simulated datasets. Example simulated data are presented in Supplementary Movies 1 and 2.

conventional fitting algorithms (see Supplementary Note 2 and Methods). Each dataset was also simulated for typical photon values expected for a fluorescent protein (1000 photons), an organic dye (4000 photons), and a photon-unlimited fluorescent probe (10,000 photons) to reflect different labeling scenarios[29]. Computational multi-emitter fitting was not implemented in the analysis to enable the direct evaluation of optical performance and ensure a fair comparison since algorithms are at different levels of technical development for each technique[21,30]. Especially since single-emitter algorithms have been shown to outperform multi-emitter algorithms in high-density 3D SMLM scenarios[12].

Direct comparison of PSF footprint with DoF in Supplementary Fig. 6 reveals how the area of each 3D PSF changes with axial position. Compared to the other techniques, SMLFM differs in that it breaks the observed trade-off trend between PSF size and DoF. This allows for an axial range that is suitable for imaging entire cells up to $8\ \mu m$, with a PSF area that is on average 55% the size of the double helix PSF (DHPSF). As every perspective view comprises a super-resolvable image of the sample, only the area of the SMLFM PSF in each perspective view is needed for direct comparison. The simple and compact PSF footprint of SMLFM is a principle component in the ability to resolve single emitters at higher spot densities than the double helix and tetrapod PSFs. In this work, we consider the central view for density studies.

Several quality-of-imaging metrics were then computed for each simulated dataset classifying a localization as either a true positive

(TP), false positive (FP), or false negative (FN) with respect to known ground-truth (GT) coordinates. The positive predictive value (PPV, also known as *precision*) describes the fraction of TP localizations relative to all localizations (TP+FP). Sensitivity (also known as *recall*) describes the fraction of accurate localizations that are retrieved (TP/GT). Both PPV and sensitivity are presented as a function of $\rho_{loc}$ in Fig. 2b using datasets simulated at 4000 detected photons to reflect labeling using an organic dye molecule (see Supplementary Fig. 9 for PPV and sensitivity plots at 1000, 4000, and 10,000 detected photons, and Supplementary Fig. 10 for associated metrics). Jaccard index (TP/TP + FP + FN) was also computed as a function of $\rho_{loc}$ and is presented in Supplementary Fig. 11.

Low signal-to-noise ratio (SNR), high background fluorescence, and emitter overlap contribute to reconstruction artefacts from the incorrect localization of single emitters. This leads to a decrease in both PPV and sensitivity as a function of $\rho_{loc}$, in agreement with similar work[6]. Unlike for the double helix or tetrapod PSFs, the PPV for SMLFM is linear across the whole $\rho_{loc}$ range with a maximum value of $100.0 \pm 0.0\%$ (mean $\pm$ SD) at $\rho_{loc} = 0.005\ \mu m^{-2}$ and a minimum of $96.7 \pm 0.1\%$ at $\rho_{loc} = 0.375\ \mu m^{-2}$, which can be rationalized by the spatiotemporally correlated PSF filtering out stochastic noise due to the requirement for the same emitter to be localized in each perspective view. PPV for the standard and astigmatic PSFs is also linear across all values of $\rho_{loc}$ as expected from compact (but very low DoF) PSF footprints. Conversely, with an average pixel area of $1.8 \times$ that of the SMLFM PSF, the DHPSF exhibits a non-linear response to $\rho_{loc}$ and a

much lower PPV than SMLFM and likewise with the tetrapod PSF. Their weaker resistance to increasing $\rho_{loc}$ can be attributed to their greater size and complexity of photon distributions. For example, the tetrapod PSF was specifically designed for optimal Fisher information, and computational multi-emitter fitting is generally implemented alongside lower density imaging scenarios[21,31].

A linear relationship between sensitivity and $\rho_{loc}$ is also observed for SMLFM with a maximum value of $100.0 \pm 0.0\%$ when $\rho_{loc} = 0.005\ \mu m^{-2}$ and a minimum of $42.5 \pm 0.1\%$ when $\rho_{loc} = 0.375\ \mu m^{-2}$. This is a result of distinguishing overlapping emitters through parallax, whereby single-molecule fluorescence is observed at different positions in each perspective view. Resolving overlapping emitters through the double helix and tetrapod PSF shaping methods is either impossible or computationally expensive during post processing. Alternatively, SMLFM facilitates these higher localization rates by resolving emitters through parallax, described herein as optical multi-emitter fitting (distinct from computational multi-emitter fitting). These data combined demonstrate that SMLFM has the capacity to localize $86.6 \pm 0.9\%$ of all emitters at a typical 2D-SMLM localization density of ~ $0.1\ \mu m^{-2}$ without compromising on the total number of localizations[32]. Even at an incredibly high localization density of $0.375\ \mu m^{-2}$ SMLFM is able to recover 43% of all ground-truth localizations while this is less than 1% for the double helix and tetrapod PSFs.

By comparing the spot densities at which the DHPSF and SMLFM achieve equal error rates we determine a maximum speed improvement of $8.95 \times$ for SMLFM at an error rate of 13.5% (at which 86.5% of all localizations are correctly reconstructed in 3D, see Supplementary Note 4 and Supplementary Fig. 8) for 4000 detected photons. This represents the upper practical limit in what SMLFM can achieve in direct comparison with the DHPSF (the state-of-the-art 3D SMLM modality for DoF and localization precision). Furthermore, this maximum practical speed improvement was measured to be $25.3 \times$ (an error rate of 40.0%) at 1000 detected photons and $10.6 \times$ (error rate of 11.0%) at 10,000 photons. Therefore, on the basis of speed, SMLFM significantly outperforms the DHPSF at all light levels, particularly at low SNR, due to optical multi-emitter fitting.

## SMLFM captures the heterogeneity of live B-cell membrane receptors

The density-dependence studies reveal an optical redundancy in SMLFM that would be suited to the high-density volumetric imaging of entire cells through optical multi-emitter fitting. To challenge our method we imaged whole primary mouse B-cell membranes in a scan-free dSTORM modality previously optimized for 2D-SMLM[33–35]. The 3D organization of membrane receptors on immune cells, such as the B-cell receptor (BCR) is of increasing scientific interest to better understand the immune response to infection[36,37]. Single BCR complexes were labeled with a single molecule of Alexa-Fluor 647 and imaged under an inclined illumination angle to improve contrast. An average of 40,000 3D localizations were accumulated per cell over an axial range of ~ 8 μm (Fig. 3a–d). Membrane ruffles and microvilli could be observed, consistent with sub-diffraction resolution being obtained[38]. For a comprehensive optical description of the SMLFM platform see Methods, Supplementary Note 1 and 3, and Supplementary Figs. 12 and 13.

3D localizations were collected over a ~ 50 μm² circular detector area (image space) with an average localization density of ~ 0.10 μm⁻² (see Supplementary Figs. 14 and 15). A maximum localization density of ~ 0.24 μm⁻² was achieved for a small portion of the experiment. Our simulations indicate that at a localization density of 0.10 μm⁻², both SMLFM and astigmatism (most commonly used) achieve equal sensitivity, $86.6 \pm 0.9\%$ and $86.4 \pm 0.5\%$ (see Fig. 2b), and a Jaccard index of $86.0 \pm 0.9\%$ and $82.6 \pm 0.6\%$, respectively (see Supplementary Fig. 11). The comparable resolving power of both techniques combined with the 8-fold larger depth-of-field afforded to SMLFM, eliminates the need

for axial scanning, and elevates SMLFM to a region of high biological throughput and applicability. Importantly, SMLFM is advantageous at these high densities because single emitters are not required to be isolated in every perspective view to be localized in 3D. We quantified this redundancy that enables optical multi-emitter fitting in these large cellular datasets by considering the localization precision as a function of a number of perspective views used for PSF fitting. Figure 3e reveals a median localization precision of <40 nm laterally and <47 nm axially and these values improved to ~ 30 nm laterally and ~ 34 nm axially as the number of perspective views for fitting was systematically increased from 3 to 7. This is consistent with a higher effective numerical aperture and better sampling of the PSF position when utilizing a greater number of perspective views. Attempts were made to ensure the distribution of localizations per number of views was equal, see Fig. 3f. Taken together, these data show that optical multi-emitter fitting via parallax is a powerful approach to 3D localizing single molecules at high densities within cells.

Another important application of the high emitter density measurements afforded by SMLFM is single-particle tracking (SPT)[39–41]. 3D-SPT better quantifies diffusive processes than 2D measurements, which tend to underestimate diffusion speed[42,43]. A previous study of membrane protein mobility highlighted the importance of imaging diffusion dynamics away from the glass interface (basal surfaces)[40], which sparked the imaging of apical surfaces in 4 μm optical sections using the DHPSF[5]. SMLFM boasts a significant practical advancement over this work for two key reasons. Firstly, the larger DoF ensures single proteins can be tracked over entire cell volumes without scanning. Secondly, localizing emitters through parallax improves the ability to delineate trajectories that would otherwise be occluded at higher densities. To demonstrate this we applied SMLFM to the 3D-SPT of BCR complexes found on the surface of live mouse B cells, accumulating hundreds of trajectories in <10,000 frames (~5 min) with an average track length of 12.5 points (Fig. 4a). Maximum likelihood estimation of the diffusion coefficient from trajectories over 5 cells (a total of 1806 tracks) yielded a distribution of diffusion coefficients (Fig. 4b–d and Supplementary Fig. 16) for individual BCR complexes with a median value of $0.14 \pm 0.08\ \mu m^2\,s^{-1}$, consistent with that observed by Tolar et al. on resting murine B cells[44]. To confirm this, we measured a median diffusion coefficient of $0.20 \pm 0.01\ \mu m^2\,s^{-1}$ at the apical surface using fluorescence correlation spectroscopy.

SMLFM effectively and accurately captures the heterogeneity of diffusion coefficients of surface receptors and opens up the possibility of the direct observation of dynamic BCR clustering following or proceeding antigen encounters (and in general the clustering of key signaling proteins in other systems), which is of great interest in the study of receptor triggering[45].

## High-density SMLFM resolves intracellular structure

To demonstrate intracellular imaging at very high emitter density, we performed scan-free dSTORM imaging of tubulin in fixed HeLa cells with SMLFM. Figure 5a shows a snapshot of raw localization data with a cell occupying a 40 μm × 40 μm FoV, which spanned an axial range of ≤3 μm, with the 3D reconstruction shown in Fig. 5b. A maximum of 40 3D localizations were detected per image frame, with an average of ~ 22 per frame, totaling 150,000 localizations over ~ 4 min (30 ms detector exposure, Fig. 5c). This corresponds to a maximum density of 0.15 μm⁻² and an average of 0.075 μm⁻².

A median value of 3922 photons was detected per 3D localization (Fig. 5d) achieving median lateral and axial localization precisions of 51.4 nm and 57.4 nm, respectively (see Supplementary Fig. 17). Line profiles (Fig. 5e, width 400 nm) were drawn for two ranges to confirm the resolution of individual microtubules. A Fourier Shell Correlation (FSC) of 59 nm resolution at a 1/7 cut-off was calculated from the localizations presented in Fig. 5b (see Supplementary Fig. 18).

 

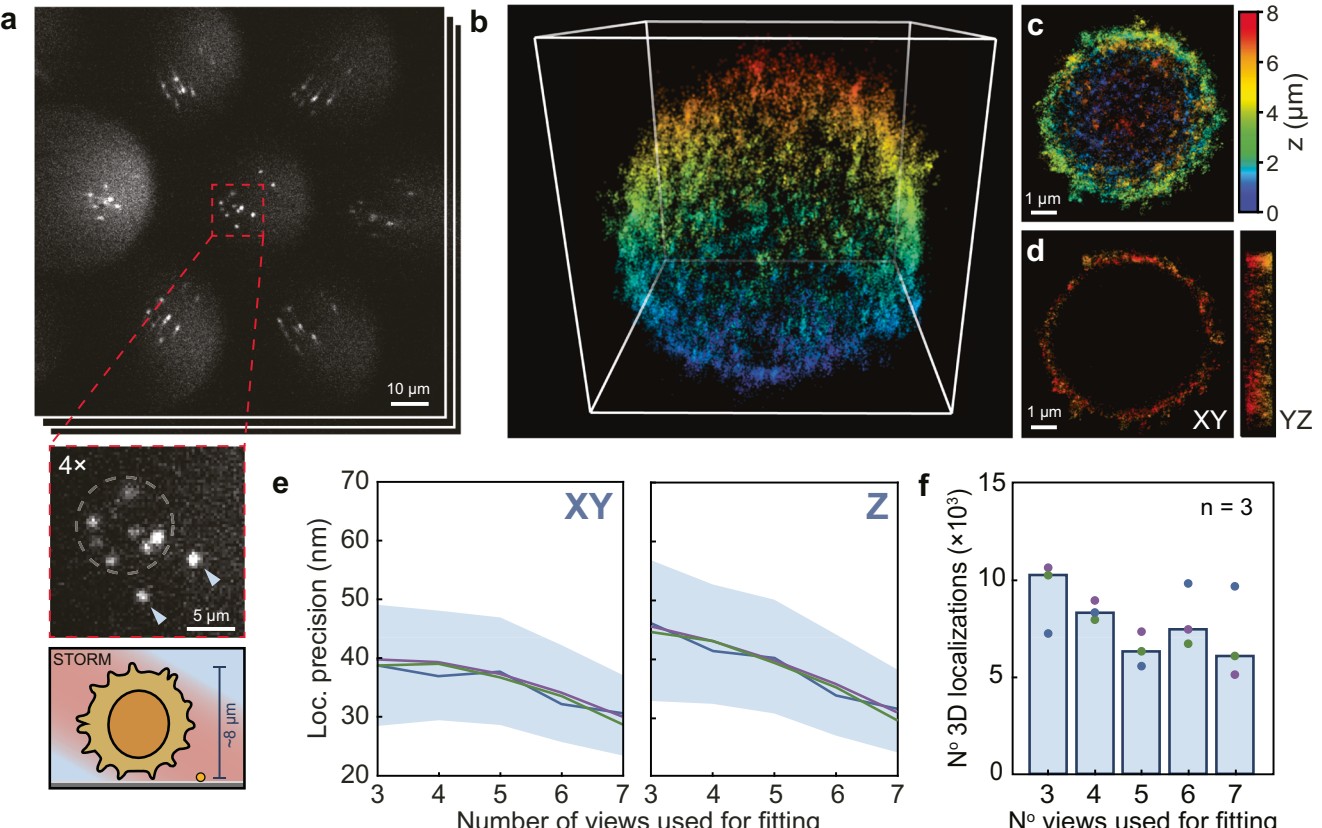

**Fig. 3 | Scan-free SMLFM-STORM imaging of B-cell receptors over whole primary mouse B cells. a** Representative frame of SMLFM localization data showing individual membrane receptors through 7 perspective views in a hexagonal arrangement (total of 50,000 frames per cell, where $n = 3$ cells). The expanded insert shows 7 fluorescent puncta (AlexaFluor 647) in the central perspective view and two fiducial markers indicated by arrows. Directly below is an illustration of the cell being imaged. **b** Associated 3D reconstruction of the whole primary mouse B cell (40,000 3D localizations in a 9 μm³ box), **c** an xy projection, and **d** a 1 μm thick central clipping to illustrate non-internalization of dye molecules. **e** Median lateral and axial localization precision (fitting error) for localizations below 60 nm resolution as a function of the number of views used to reconstruct a 3D localization (shading represents interquartile range). **f** Proportion of 3D localizations below 60 nm lateral precision as a function of the number of views used to reconstruct in 3D. A total number of 3D localizations per view is represented by dots (where $n = 3$ cells), with the mean value represented by each bar. Experimental data and the accompanying real-time 3D reconstruction are presented in Supplementary Movie 3.

Furthermore, no 3D-specific sample optimization was undertaken prior to imaging, and a dSTORM buffer protocol was implemented that was previously developed for 2D-SMLM[34]. As protocol optimization for SMLM can be time-intensive and challenging, this facile translation from 2D to 3D SMLM presents a significant advantage in the future use of 3D-SMLM in biological research[46].

In summary, we reported hexagonal SMLFM and quantitatively compared its performance to other 3D SMLM approaches. SMLFM enables an order-of-magnitude imaging speed improvement compared to DHPSF microscopy, which we attribute to optical multi-emitter fitting by which overlapping emitters can be resolved through a redundancy in the number of perspective views required for 3D reconstruction. We illustrate this redundancy experimentally by imaging both live and fixed whole cells and dense arrays of cytosolic tubulin. Specifically, utilizing fewer than all 7 views for 3D fitting has minimal impact on the resulting localization precision, which commonly occurs in dense emitter datasets.

Future endeavors could couple SMLFM with computational multi-emitter fitting and/or deep learning strategies, high-speed detectors, and alternative volumetric labeling strategies to push localization rates even further. We anticipate SMLFM to be a valuable tool in improving our understanding of three-dimensional nano-scale architectures and dynamics, and a key step towards real-time 3D super-resolution imaging in the life sciences.

# Methods

## Ethical statement

The organism used in this study (C57BL/6J mouse) was sacrificed under Schedule 1 conditions following the regulations set by the UK Home Office. Spleens were harvested ex vivo and B cells were isolated from splenocytes. No ethics approval was required for this study.

## Optical setup

The SMLFM platform described in this work was constructed using an epi-fluorescence microscope (Eclipse Ti-U, Nikon) housing a 1.27 NA water immersion objective lens (Plan Apo VC 60 × , Nikon, Tokyo, Japan) for imaging above the coverslip. The z-position of the objective was controlled with a scanning piezo (P-726 PIFOC, PI, Karlsruhe, Germany). The Fourier lens ($f = 175$ mm, ThorLabs) was placed in a 4f configuration with the tube lens ($f = 200$ mm, Nikon) to relay the back focal plane (BFP) outside of the microscope body (see Supplementary Fig. 1). A hexagonal microlens array ($f = 175$ mm, pitch = 2.39 mm, custom-made by CAIRN Research) was placed in the BFP to relay the image plane onto an EMCCD (Evolve Delta 512, Photometrics, Tucson, AZ, 16 μm pixel size). Excitation was achieved using a 640 nm ( ∼ 10 kW cm⁻² power density, 150 mW, iBeam Smart-S 640-S, Toptica, Munich, Germany) and activation by a 405 nm ( ∼ 0.04 kW cm⁻² power density, 120 mW, iBeam Smart-S 405-S, Toptica, Munich, Germany) laser, that were circularly polarized, collimated and focused on to the

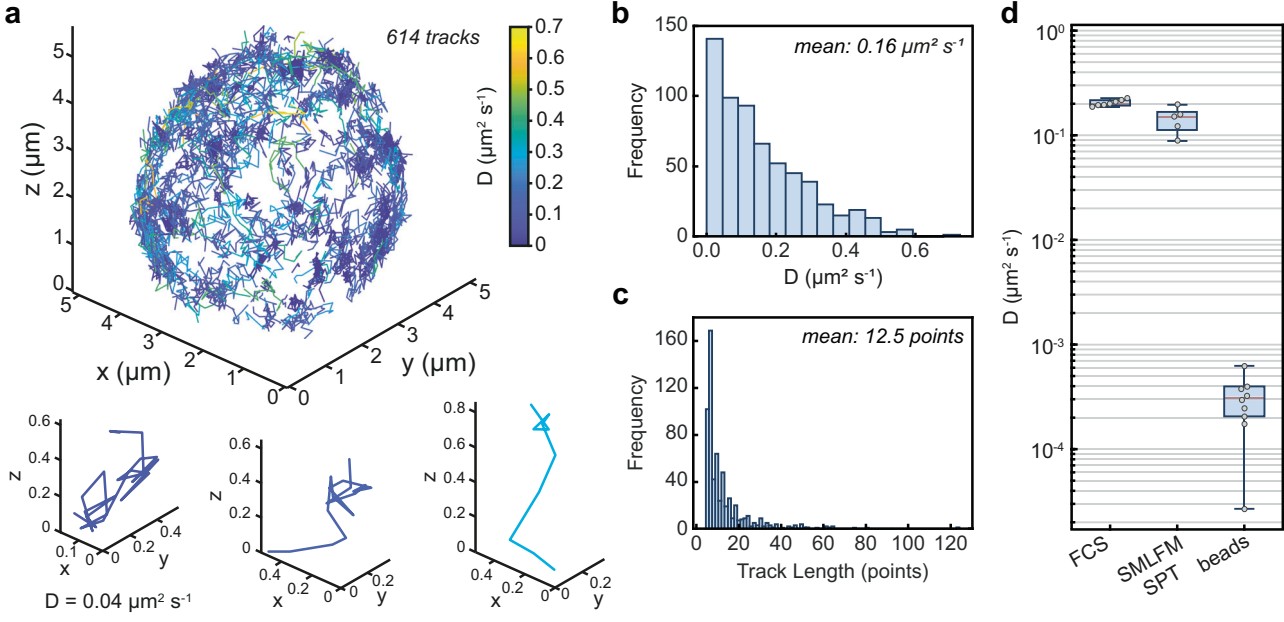

**Fig. 4 | Scan-free whole-cell 3D SPT of the B-cell receptor on primary mouse B-cell membranes using SMLFM. a** 3D trajectory map of the BCR over a whole primary mouse B-cell totaling 614 tracks color-coded by diffusion coefficient using maximum likelihood estimation. Example isolated trajectories of varying diffusion coefficients are expanded directly below. Histograms showing **b** individual diffusion coefficients, and **c** track lengths, from the cell presented in **a** (bin widths were determined using Freedman-Diaconis' rule). **d** Diffusion coefficient of the BCR measured by FCS ($n$ = 7 cells) and SMLFM-SPT ($n$ = 5 cells), and the minimum diffusion coefficient measurable by SMLFM-SPT determined with immobilized beads ($n$ = 9 regions). The center of each box represents the median average diffusion coefficient over $n$ repeats (data points from repeats overlaid), with the box bounds representing the 25th and 75th percentile, and the whiskers representing maximum and minimum values. SMLFM-SPT comprises a total of 1806 trajectories over 5 cells with a mean track length of 8 points. Example experimental data and reconstructed 3D trajectories are presented in Supplementary Movie 4.

BFP of the objective. Unless stated otherwise, samples were excited with a highly inclined and laminated optical sheet (HILO) which was achieved by laterally displacing the excitation beam towards the edge of the BFP of the objective (see Supplementary Note 1.3). Fluorescence was collected by the same objective and separated from the excitation beam using a quad-band dichroic mirror (Di01-R405/488/561/635-25 × 36, Semrock, Rochester, NY). Long-pass (BLP01-647R-25, Semrock) and band-pass (FF02-675/67-25, Semrock) emission filters were placed immediately before the detector to isolate fluorescence emission. The pixel size in image space was measured at 266 nm.

### 3D reconstruction of SMLFM data
All experimental data were recorded as .tif stacks. 2D Gaussian fitting of all emitter positions in all perspective views was carried out in Fiji using PeakFit (GDSC SMLM 2.0) to yield a set of 2D localizations for each raw frame. Given this initial set of 2D localizations, individual emitters were localized in 3D using custom Matlab scripts available at ref. 47 as outlined in ref. 22. Briefly, the most likely subset of 2D localizations in different perspective views corresponding to a unique emitter was identified. Provided that this set of localizations contained more than 3 elements, the 3D location of this emitter was calculated as the least-squares estimate to an optical model relating the axial emitter position to the parallax between perspective views. If the residual light field fit error was below 200 nm, the fit was accepted and the subset of 2D localizations was removed. This procedure was repeated for each individual emitter. Drift correction was performed by localizing the position of a fiducial marker in each frame and subtracting the resulting 3D fiducial points from all localizations of the corresponding frame. System and sample aberrations were corrected by subtracting the residual disparity (calculated for data acquired for all emitters localized during the first 1000 frames) from all 2D localizations prior to calculating the 3D light field fit. For full details of the light field

localization fitting procedure refer to the Supplementary Information of ref. 22. All experimental 3D dSTORM data using AF647 underwent temporal grouping using bespoke code to compensate for single-molecule fluorescence extending beyond a single frame (Supplementary Fig. 14). Sequential localizations, within a 4-frame interval, were deemed to originate from the same molecule if they appeared within a 3D volume dictated by the localization precision. The precision (xyz) and detected photons for the brightest localization within this subset were kept and all other localizations were removed from the subset. 3D visualization was carried out in ViSP[48].

### Optical 3D calibration
Fluorescent beads (200 nm, Deep Red FluoSpheres, ThermoFisher, Waltham, MA) were immobilized on a glass slide and imaged to calibrate for deviations in experimental and calculated the disparity from the SMLFM optical model. Glass slides were cleaned under argon plasma (PDC-002, Harrick Plasma, Ithaca, NY) for 1 h and incubated with poly-L-lysine (PLL, 50 μL, 0.1% w/v, Sigma-Aldrich, P820) for 10 min. Glass slides were washed with PBS (3 × 50 μL) and incubated with fluorescent beads (50 μL, ca. $3.6 \times 10^8$ particles/mL) incubated for 3 min before washing further with PBS (3 × 50 μL). The piezo stage (P-726 PIFOC, PI, Karlsruhe, Germany) was used to scan the objective lens axially over 8 μm recording 10 frames at 30 ms exposure per 60 nm increment. The data was reconstructed in 3D and plotted against the known movement of the piezo stage. A linear fit was applied to the calibration curve, the gradient of which was a correction factor subsequently applied to all reconstructed data presented in this work.

### SPT analysis
Following 3D reconstruction of SMLFM data, a custom-written MATLAB code was implemented to temporally group localizations into single trajectories[47]. Some parameters were chosen by the user, including a

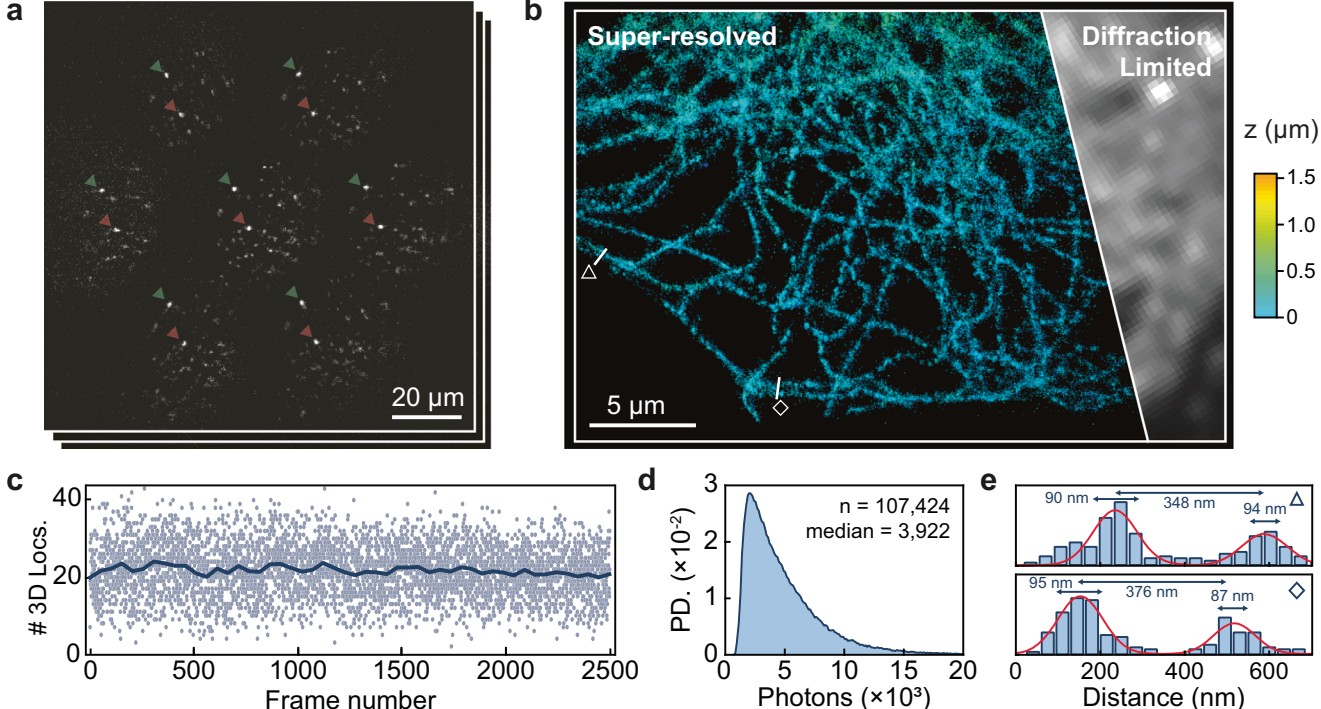

**Fig. 5 | dSTORM imaging of AlexaFluor 647-labeled tubulin in a HeLa cell.** **a** Representative frame of SMLFM localization data showing microtubule-stained HeLa cells through 7 perspective views in a hexagonal arrangement (total of 40,000 frames, imaged to completion). Two fiducial markers are indicated by arrows. **b** The corresponding super-resolved 3D volume, containing 173,314 localizations, color-coded by depth. **c** Localization rate over the first 2500 frames indicating a mean 3D localization rate of ~ 22 frame⁻¹ (blue line, rolling average over 100 frames) and an upper limit of ~ 40 frame⁻¹, corresponding to a localization density of ~ 0.075 and ~ 0.15 locs μm⁻², respectively. **d** Histogram of detected photons per 3D localization. **e** Line plots of width of 400 nm illustrate the resolution of individual microtubules as indicated by the triangle and diamond in **b**.

number of dark frames, linking distance, and minimum track length. The diffusion coefficient was then calculated from each trajectory using maximum likelihood estimation, which has previously been shown to yield statistically robust measurements of the diffusion coefficient[49].

To determine the minimum observable diffusion coefficient, fluorescent beads (200 nm, Deep Red FluoSpheres, ThermoFisher, Waltham, MA) were immobilized on a glass slide and imaged under conditions (641 nm excitation at ~ 2 kW cm⁻² power density, 20 ms exposure time) that artificially reproduce the same photon intensities as PA-JF646 used for SPT experiments. The raw data was reconstructed in 3D and trajectories were analyzed as described earlier in Methods to yield the smallest resolvable diffusion coefficient.

**Analysis of simulated data**
2D and astigmatic datasets were fitted in PeakFit (GDSC SMLM 2.0, Fiji plug-in) using a circular and astigmatic Gaussian PSF, respectively. DHPSF datasets were initially fitted in PeakFit using a circular Gaussian PSF before 3D reconstruction using DHPSFU. SMLFM datasets were initially fitted in PeakFit using a circular Gaussian PSF before 3D reconstruction using a custom MATLAB code described earlier in Methods. Tetrapod PSF data was fitted using Zola (Fiji plug-in) for 3D reconstruction[50].

A custom MATLAB code[47] was written to compare the fitted (3D, 2D for the standard PSF) point data to the ground-truth coordinates. Specifically, the root mean square distance matrix is calculated between all ground-truth coordinates and all reconstructed data points on a frame-by-frame basis and counted as either a true positive, false positive or false negative given a user-specified distance tolerance. The tolerance applied was different for each technique and dictated by the precision and thresholds (determined by the fitting error) were applied to determine true positives and false positives.

**Preparation of coverslips for B-cell imaging**
Glass slides (VWR, 631–1570) were washed with propan-2-ol and water, dried under nitrogen and cleaned under argon plasma (PDC-002, Harrick Plasma, Ithaca, NY) for 1 h. Glass slides were then incubated with poly-ʟ-lysine (PLL, 50 μL, 0.1% w/v, Sigma-Aldrich, P820) for 1 h and washed with filtered (0.02 μm syringe filter, Whatman, 6809-1102) PBS (3 × 50 μL) before incubation with gold nanoparticles (5 μL, 0.1 μm, Merck) for 20 min.

For fixed cell imaging, glass slides were then washed with filtered PBS (3 × 50 μL). 1 × 10⁵ fixed labeled B cells were washed in dSTORM buffer (50 mM Tris-HCl, 10 mM NaCl, 10% glucose, 10 mM MEA, 84 μg mL⁻¹ catalase, 0.2 mg mL⁻¹ GLOX, adjusted to pH 8), plated in 20 μL dSTORM buffer and left to settle for > 20 min. Prior to imaging, the sample was washed into fresh buffer dSTORM buffer.

For live cell tracking, PLL-coated glass slides were prepared as above and placed in filtered PBS. For SPT, the surface was incubated with gold beads as above, washed 3 × in filtered PBS, and cells labeled with PA-JF646-conjugated Fab-Halo were allowed to settle onto the surface for 5–10 min prior to imaging.

For point fluorescence correlation microscopy (pFCS), cells labeled with AF647-conjugated Fab-HaloTag were incubated onto the PLL surface for 5–10 min and imaged using a Zeiss LSM780 inverted confocal microscope using a 40 × water objective, with the sample excited using a 633 nm He-Ne laser. The confocal volume was placed on the apical surface of the cell membrane and five repeated measurements were taken per cell. Data was analyzed using PyCorrFit and the diffusion coefficient was calculated from the average transit time (τ), using the confocal beam width as calculated using a solution of 100 nM AF647 HaloTag ligand solution.

## B-cell culture and fluorescent labeling

Primary murine B cells were isolated from the spleens of male C57BL/6J mice aged between 8 and 12 weeks. Splenocytes were isolated by mechanical disruption of the spleen, and incubated with ACK lysing buffer (Lonza, LZ10-548E) for 2 min at room temperature to lyse erythrocytes. The cells were washed in RPMI-1640 (Gibco) medium supplemented with 10% fetal bovine serum (FBS) and B cells were isolated using a B-Cell Isolation Kit, mouse (Miltenyi Biotec, 130-090-862) according to the manufacturer's instructions. Purified murine B cells were either resuspended in PBS for dSTORM labeling or frozen in FBS supplemented with 10% DMSO to later culture for live cell imaging (SPT and FCS).

BCR complexes were labeled using a recombinant protein based on the Fab fragment of the anti-murine CD79b antibody HM79-16. A self-labeling HaloTag domain was introduced to the C-terminus of the Fab heavy chain to ensure single-dye labeling of the probe. Fab-Halo protein was labeled with HaloTag ligand dyes by incubation with a 2-fold molar excess of dye for 90 min at room temperature, with free dye removed using a Bio-Spin P-6 gel column (BioRad, 7326227) according to manufacturer's instructions. The labeled protein was aliquoted and stored at $-80\,°C$. For dSTORM imaging, freshly isolated C57BL/6J B cells were labeled at $4\,°C$ with recombinant AlexaFluor 647 Fab-Halo protein. $2 \times 10^6$ cells were washed in 0.22 µm filtered PBS and incubated in 2.5 µM Fab-Halo (AF647) for 45 min at $4\,°C$. Cells were washed twice in cold filtered PBS, fixed in 1% paraformaldehyde (Sigma, 28906) for 30 min at $4\,°C$, and placed in filtered PBS at a final density of $4 \times 10^7$ cells mL$^{-1}$.

For live-cell imaging, as conducted for SPT and FCS, cells were thawed from frozen stocks and cultured in primary B-cell medium (RPMI-1640 supplemented with 10% FBS, 2 mM L-Glutamine, 10 mM HEPES, 1 mM sodium pyruvate, 50 µM 2-mercaptoethanol, 50 U mL$^{-1}$ penicillin and 50 µg mL$^{-1}$ streptomycin), supplemented with 10 µg mL$^{-1}$ anti-mouse CD40 (clone 1C10, Biolegend 102812) and 10 ng mL$^{-1}$ murine IL-4 (Peprotech, 214-14). For live cell imaging, $2 \times 10^5$ cells were washed in filtered PBS and incubated with 1 µM fluorescent Fab-Halo for 15 min at room temperature and washed twice in PBS prior to incubation with the coverslip.

## HeLa cell culture and fluorescent labeling

HeLa cells (TDS, Dresden, Germany) were cultured at $37\,°C$ and 5% $CO_2$ in DMEM (Gibco, Invitrogen) supplemented with 10% FBS (Life Technologies), 1% penicillin/streptomycin (Life Technologies), and 1% glutamine (Life Technologies). Cells were passaged every three days and were regularly tested for mycoplasma. One day prior to fixation, cells were seeded on high-precision 1.5 glass coverslips (MatTek, P35G-0.170-14-C) for imaging.

Cells were fixed and permeabilized simultaneously for 6 min in Cytoskeleton Buffer with Sucrose (CBS, 10 mM MES, 138 mM KCl, 3 mM $MgCl_2$, 2 mM EGTA, and 4.5% sucrose w/v, pH 7.4) containing 4% methanol-free formaldehyde (FA, Fisher Scientific) and 0.2% Triton, followed by a second fixation for 14 min in CBS + 4% methanol-free formaldehyde at $37\,°C$ and 5% $CO_2$. Post fixation, cells were washed three times in PBS + 0.1% Tween (PBST), and further permeabilised in PBS + 0.5% Triton for 5 min. Cells were then washed in PBST three times and blocked in 5% BSA (in PBS) for 1 h at room temperature. Samples were further washed three times in PBST, after which samples were incubated with an anti-α-tubulin antibody (ab7291, clone DM1A, at 2.5 µg mL$^{-1}$ in 5% non-fat milk) overnight at $4\,°C$. Cells were then washed six times in PBST after which a Donkey anti-Mouse IgG (H+L) Highly Cross-Adsorbed Secondary Antibody AlexaFluor 647 (Invitrogen, A-31571, at 2.0 µg mL$^{-1}$ in 5% non-fat milk) was added to the sample for 1 h at $4\,°C$. The cells were then washed six times in PBS and the sample flooded with STORM imaging buffer prepared as described earlier in Methods.

## Statistics and reproducibility

No statistical analyses were performed in this paper and no statistical methods were used to predetermine sample size. Where relevant, sample sizes are reported in the figure legends, and all experiments were performed independently at least three times to ensure results were repeatable.

## Reporting summary

Further information on research design is available in the Nature Portfolio Reporting Summary linked to this article.

## Data availability

All B-cell receptor (live and fixed) and tubulin data generated in this study (accompanying Figs. 3–5) have been deposited on *Zenodo* [https://doi.org/10.5281/zenodo.8190164][47]. Source data for Figs. 2–5 are provided with this paper. Source data are provided in this paper.

## Code availability

All analysis code and example data used in the preparation of this work −including 3D reconstruction code, 3D tracking code, and ground-truth matching code−are available via Zenodo (https://doi.org/10.5281/zenodo.8190164)[47]. A managed version of the hexagonal SMLFM reconstruction code is also available on GitHub at https://github.com/TheLeeLab/hexSMLFM.

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

## Acknowledgements

This work was supported by The Royal Society grant (SFL, RGF/EA/181021). We would like to thank Jeremy Graham at CAIRN Research for providing the hexagonal microlens array, Alexander Collins for useful discussions, and Gregory Chant and James McColl for optimizing the dSTORM buffer. We also thank Janelia Materials for providing the PA-JF646 Halo-Tag used for 3D-SPT.

## Author contributions

S.D., A.H., and S.F.L. conceived the project. S.D., J.F.F., S.F.L., S.J.D., D.K., and K.O.H. designed experiments. S.F.L. supervised the research. S.D. and A.H. built the optical set-up and performed baseline experiments. S.D. performed all imaging and data analysis. J.F.F. prepared B-cell samples and conducted FCS experiments. E.B. programmed and performed the simulations. R.P. and B.Z. prepared tubulin samples. ES provided samples for early tests. R.R.S., K.O.H., S.B., and B.Z. wrote and maintained the 3D reconstruction code. JSB provided a diffusion analysis code. S.D. and S.F.L. wrote the manuscript with input from all authors.

## Competing interests
CAIRN Research has a co-development agreement with S.F.L. and K.O.H. at the University of Cambridge. The remaining authors declare no competing interests.
