## [Peer Review File · Nature Communications]

REVIEWERS' COMMENTS

Reviewer #1 (Remarks to the Author)

The document can be divided into two different, almost independent parts. In the first part, the paper reports a very interesting study comparing the performances of different microscopy techniques when used for the single-molecule tracking task.

The study is carried out computationally. Its implementation is rigorous, and its results can be of great help to researchers interested in tracking a single molecule in 3D. However, by itself, this part does not merit publication in Nature Communications, as it does not contribute new science.

More serious are my concerns about the second part of the article in which, as the authors themselves state, they report "the first hexagonal SMLFM platform that enables 3D-SMLM in an axial range of 8 μm ". From my point of view, this is the poorest part of the job, for the following reasons:

1.- I don't see anything new here. Hexagonal microlenses are widely used in Fourier light field (see articles published by the research groups writing references [23] and [24] of the manuscript). In fact, the system configuration in this article is quite similar, apart from the hexagonal geometry, to the one reported by the same authors in their OPTICA article.

2.- The proposed optical setup is neither well designed nor well implemented. On the one hand, the use of very large focal lengths (both for the Fourier lens and for the microlenses) results in a very long add-on (I calculate 525 mm) and a very small FoV (40 μm). Surely other authors (or even the same authors) have proposed better designs.

The optical implementation is the poorer side of the article. For example, if simple calculations are carried out, a theoretical value of $N=2.4$ is obtained for the described optical configuration. However, the authors show experimental images with $N=3$. But this should not be possible, unless the microlenses were projected not at the stop aperture, but very far from that position. In such a case, more than 2.4 microlenses could receive light from the sample, but with very strong vignetting (as can be seen in the Supplementary Movies 2, 3 and 4). Another problem that I detect in the images is the strong overlap between the views (this is very apparent in images 426778_0_data_set_7579143_rts8kj.tif). This should never happen and could be avoided by inserting the proper field stop.

All these optical mismatches cause the system to provide much poorer perspective views than the Fourier lightfield can provide. Therefore, the results in single-molecule tracking could be much more impressive than those reported here. Therefore, I encourage the authors to work further on the optical implementation, which will allow them to provide much more useful results to the research community.

3.- The reconstruction algorithms do not seem to be new, but rather the same (or slightly modified) as others previously reported by the same authors.

In summary, the theoretical comparison between different single-molecule tracking techniques is interesting and rigorous, but I don't think it provides enough new science to merit publication in Nature Communications. On the other hand, the implementation of "high-density volumetric super-resolution microscopy" does not seem to contribute new science and is very poor since the systems are not optimally designed and implemented.

Based on all these facts and ideas, I am afraid I cannot recommend acceptance of this article.

Reviewer #2 (Remarks to the Author)

The manuscript "High-density volumetric super-resolution microscopy" by Daly et al. focuses on the quantification of accessible emitter densities in single-molecule light field microscopy (SMLFM). In SMLFM, an array of micro lenses (MLA) is placed in the Fourier plane (conjugated back-focal plane) creating a pattern of PSFs that are then recorded via a camera and analysed using standard (for PSF localisation) and custom (for connecting the localisations from the different areas) software. The work is a logical extension of the author's previous work (Sims et al., reference 22), in which the concept and the ability to image an impressive z-range of 8 μm had been demonstrated. Here, the authors used a customised MLA with a hexagonal arrangement improving the usable area on the camera. In addition to the comparison with other methodologies enabling 3D resolution in SMLM, the authors further demonstrate successful single-particle tracking over an extended biological object (B cell receptors).

The manuscript is generally well written but could, in my opinion, be considerably sharpened to provide necessary information and additional clarity to the reader. As of now, I feel that some of the comparisons with other methods are not carried out as stringent as I would have liked to see them. Overall, I still consider the manuscript a very valuable addition to the field and I hope that the authors will find the following points useful to improve the manuscript.

Major points

- Figure 1 a/b): If I am not mistaken, the work uses a 4f arrangement of lenses as shown in one of the SI figures and not the arrangement shown here. I would suggest to merge the SI figure and the current Figure 1.
- Figure 1/2: I find it rather confusing to see different pixel sizes for different modalities in, apparently, the same field of views (in terms of constant area?). Is the reason for this a different effective magnification of the optical system? This choice makes comparing achievable emitter densities on a given camera frame area (!) rather difficult. (see also next point)
- Figure 2a: If the FoV stays the same (10x10 μm), the different binning would mean that even in a standard configuration 4x less molecules could be placed when a larger pixel size is used, correct? Is Fig 2b correcting/normalising for the different magnification? It is surprising to see Light Field doing so well compared to Standard with the 4x lower number of pixels. (The focal length of both Fourier lens or MLA was given as 175mm so I would not have expected any changes in magnification, correct?)
- Figure 2b: The most widely used quality of imaging metric is the Jaccard index defined as $JI = TP / (TP + FP + FN)$ (see, e.g. reference 6 and 12). Is there a particular reason to use precision (PPV) and sensitivity instead? If not, the authors might consider replacing sensitivity and precision with the Jaccard index to increase the overall readability of the entire manuscript.
- Figure 2: The authors cite the work on Tetrapods by the Shechtman lab (Nehme et al, reference 6), but somewhat hide the fact that the particular publication explicitly discusses how largely overlapping PSFs can be retrieved by their deep learning approach (DeepSTORM3D). For a truly fair comparison with the state of the art, tetrapods should be analysed with the best possible software, especially keeping in mind that the authors here, also use customised software to map

the localisations. It might well be, that DeepSTORM3D/DECODE on tetrapods is comparable in the achievable density without taking much away from the novelty and simplicity of the approach demonstrated here with the MLA.

- Figure 5: One thing that is not yet discussed is the "real-estate" on the camera. As the camera sensor is now covering multiple field of views when using MLA, the achievable "throughput" is lower than in standard configuration. Could the authors comment on that?
- line 129. photon numbers: 1000 photons (and not counts, I assume) per frame for a fluorescent protein is very optimistic number to get for longer than one frame (see also next point). Also, can the authors define what they mean by "next-generation fluorescent probe"?
- Figure 4: Tracking over such a large sample is impressive! With an average track length of 12 localisations using a very good probe, SMLFM seems rather photon hungry. So how does the approach fares with photon numbers of well below 1000 photons per localisation?
- No comments were made on the availability of the customised Matlab scripts for combining the localisation data from the different FoVs. Ideally, the software will be made available on Github (or is already).
- No comments were made on the availability of the customised MLA. Can the MLA be bought from CAIRN? Is there an estimated sales price?

Minor points

- both abbreviation are present in the abstract: SMLFM and SMFLM. Please check for typos.
- Non of the figures of the SI were numbered in the final compiled PDF that I got. (I later saw that the SI itself shows everything correctly)
- Figure 1a: unit for density is missing
- line 125: 20x20 μm mentioned here, figure shows 10x10 μm , so 0.5 locs per 10x10 I assume?
- line 220-222: phrasing. I somewhat doubt that the division of background photons has a larger effect than adding up the camera noise contributions from adding up seven field of views. Please comment.
- I might have missed that, how was the FSC calculated (software)? 3D, as in shell, correct?
- line 565: What do the authors mean by creating an evanescence wave? If I am not mistaken, TIRF would require an oil immersion objective with an NA of >1.4.
- I am surprised to see the mentioning of #1 cover slips. In most cases, the objectives are designed for #1.5 to minimise aberrations etc. Worth checking.
- line 575 unit missing
- how "easy" is it to adapt the software/noise model for sCMOS rather than emCCD?

Reviewer #3 (Remarks to the Author)

Daly et al present a new approach to volumetric SMLM imaging, splitting the light according to its position in the back focal plane and thereby forming multiple images, each taken at a different angle. This is a potentially very useful new approach; while many solutions have been suggested to the challenge of how to discriminate axial position, astigmatism is still the most commonly used method despite its limitations (limited z depth in a single slice, degradation of x or y precision). In particular, the authors demonstrate that their method allows use of substantially higher excitation densities than can typically be used in 3D imaging. Overall this paper represents an important and interesting new technique that could help uncover exciting new biology. However, I have some comments that I would like to be addressed.

While the acquired data is impressive the imaging of B cell receptors have a key limitation. Given the size of the cell and the number of molecules localised, I estimate that if the molecules are evenly spaced they are around 70nm apart. They are of course not evenly distributed, but it illustrates that the resolution is likely to be limited in at least some areas by the sampling rather than localisation precision (a common problem when imaging with dSTORM in thick samples). The acquired density of molecules can be limited if only a small amount of the labelled molecule is present, or by the labelling efficiency, the dSTORM blinking degrading, or the acquisition time: the authors should comment on what limited the reconstructed density in this case.

I generally think it is important to benchmark against a known structure in an experiment when

presenting new techniques. However in this case, as the focus is on accurate recall at high density rather than resolution per se, I think that the evaluation is satisfactory without additional experiments.

Minor concerns

- 1) With regard to the double helix point spread function, a brief mention of the tradeoff between PSF size and DoF for this particular technique would be helpful.
- 2) "this work will focus on techniques that yield sub-diffraction axial precision over extended axial ranges" – it needs to be clarified that the authors mean single shot techniques, stitching together multiple reconstructed planes is a standard approach to this problem.
- 3) In Figure 3a the standard and astigmatic curves look flat but I think would in fact vary. Could these curves also be shown with the vertical axis expanded so this can be seen?
- 4) In Figure 2 I found the labels giving the DoF in panel B confusing, I think the figure would be clearer with them removed.
- 5) In Figure S5 the dark grey and black lines cannot be clearly distinguished, the lines should either be colour or separated into individual graphs.
- 6) I don't think the PPV/sensitivity values are particularly helpful for the experimental data since this is from simulation. The expected high performance of the technique has been made clear in the part of the paper covering the simulations.
- 7) The FSC measurement seems lower than I would expect (while the figure may seem quite high for SMLM accurate FSC measurements tend to come out substantially higher than other resolution metrics), probably due to the calculation not taking into account multiple localisations of the same fluorophore. This is a known challenge in dSTORM which can bias measurements, and if it has not been corrected for then a note (e.g. no correction for fluorophore reappearance) should clarify that.
- 8) In the image of the microtubules the upper part of the super-resolution image appears substantially degraded compared to the lower part. Why is this? Was it due to issues keeping the excitation density low in this area? Or was this at the edge of the imaging area and this caused a degradation in quality?

RESPONSE TO REVIEWERS' COMMENTS

General remarks to reviewers are in blue; changes to the manuscript are in italicised blue.

Reviewer #1 (Remarks to the Author)

(Also see attached document)

The document can be divided into two different, almost independent parts. In the first part, the paper reports a very interesting study comparing the performances of different microscopy techniques when used for the single-molecule tracking task.

The study is carried out computationally. Its implementation is rigorous, and its results can be of great help to researchers interested in tracking a single molecule in 3D. However, by itself, this part does not merit publication in Nature Communications, as it does not contribute new science.

We thank the reviewer for their time and comments recognising the relevance of our work to help researchers.

More serious are my concerns about the second part of the article in which, as the authors themselves state, they report "the first hexagonal SMLFM platform that enables 3D-SMLM in an axial range of 8 μm ". From my point of view, this is the poorest part of the job, for the following reasons:

1. I don't see anything new here. Hexagonal microlenses are widely used in Fourier light field (see articles published by the research groups writing references [23] and [24] of the manuscript). In fact, the system configuration in this article is quite similar, apart from the hexagonal geometry, to the one reported by the same authors in their OPTICA article.

We appreciate the comment and have amended the text to be clearer and avoid misunderstanding.

The key conceptual advance in our paper was a high density 3D super-resolution technique that implements the principles of Fourier light field microscopy, we achieved this with a (yet unpublished) hexagonal MLA. This novel implementation required the development of a bespoke MLA and a new computational approach to light field fitting (available here: <https://github.com/TheLeeLab/hexSMLFM>).

We have therefore amended the text as follows to clarify:

"We report the first implementation of single-snapshot 3D super-resolution imaging over an 8 μm DoF using a hexagonal MLA."

The proposed optical setup is neither well designed nor well implemented. On the one hand, the use of very large focal lengths (both for the Fourier lens and for the microlenses) results in a very long add-on (I calculate 525 mm) and a very small FoV (40 μm). Surely other authors (or even the same authors) have proposed better designs.

We apologise for the confusion here, but we politely disagree with the comments about design. As volumetric super-resolution experiments require much higher laser fluence than, for example, diffraction limited FLFM, a FoV of 40 μm is typical (see <https://doi.org/10.1002/anie.202206919> or <https://doi.org/10.1038/s41467-017-02563-4>). We would argue that a 'fine-tuned optical optimization' is not a major goal of the manuscript, which is to demonstrate and quantify high-density 3D-SMLM. Regarding the physical size, this is in line with other single-molecule fluorescence microscopes. We used long focal length lenses to minimise field curvature and not for spatial requirements.

The optical implementation is the poorer side of the article. For example, if simple calculations are carried out, a theoretical value of $N=2.4$ is obtained for the described optical configuration. However, the authors show experimental images with $N=3$. But this should not be possible, unless the microlenses were projected not at the stop aperture, but very far from that position. In such a case, more than 2.4 microlenses could receive light from the sample, but with very strong vignetting (as can be seen in the Supplementary Movies 2, 3 and 4).

We agree that the microscope platform could be better characterised, so we have populated the Supplementary Information with optical quantities and characterising figures. The updated Table S1 details the physical parameters of the microscope platform in this study. The bespoke MLA was designed to fit the BFP, and to illustrate that all 7 lenses of the MLA are fully illuminated (giving $N = 3$) we used a camera to locate and image the back focal plane, shown in Fig. S3. The BFP diameter was measured at 7.3 mm, which taken with the MLA pitch (which is equal to double the inradius) at 2.39 mm, gives $N = 3.05$.

$N = \text{diameter of BFP} / \text{pitch} = 7.4 \text{ mm} / 2.39 \text{ mm} = 3.10$ (theoretically calculated)

$N = 7.3 \text{ mm} / 2.39 \text{ mm} = 3.05$ (experimentally measured)

Another problem that I detect in the images is the strong overlap between the views (this is very apparent in images 426778_0_data_set_7579143_rts8kj.tif). This should never happen and could be avoided by inserting the proper field stop.

While we agree that full dilation of the iris at the focal plane combined with highly inclined illumination leads to non-uniform background signal, this has no effect on the resulting localisation precision. This is illustrated by the resolution of individual microtubules in Fig 5, where the super-resolved images are consistent with the state of the art in the field (see <https://doi.org/10.1002/cphc.201300880> and <https://doi.org/10.1017/S1431927620018620>).

All these optical mismatches cause the system to provide much poorer perspective views than the Fourier lightfield can provide. Therefore, the results in single-molecule tracking could be much more impressive than those reported here. Therefore, I encourage the authors to work further on the optical implementation, which will allow them to provide much more useful results to the research community.

We recognise that many parameters are suitable to for finer optimisation in FLM optical set-ups, however the results obtained agree with all literature values (fixed cell imaging: <https://www.nature.com/articles/s41467-017-02563-4> and SPT: <https://doi.org/10.1016/j.bpj.2017.02.023>).

To avoid ambiguity, we have significantly expanded the optical characterisation of the SMLFM platform in the SI, in particular by adding Fig. S11, which is a precision curve at specific photon fluences. The curve illustrates near isotropic (xyz) localisation precision below 30 nm for an output of 2,500 photons per molecule, improving to <20 nm for 4,000 photons--typical for a dye like AF647. Altogether, this illustrates the strong resolvability of SMLFM at high density with exceptional sub-diffraction resolution.

2. The reconstruction algorithms do not seem to be new, but rather the same (or slightly modified) as others previously reported by the same authors.

The reviewer raises an important point that further development is needed in the optimal analysis pipeline, but we reserve this for a future, more focussed, study.

We have however made all code available for the first time to enable others to implement SMLFM in their work; see both Zenodo (<https://doi.org/10.5281/zenodo.8190164>) and GitHub (<https://github.com/TheLeeLab/hexSMLFM>). The fitting code had to be adapted to facilitate both hexagonal and square arrays in this work for futureproofing.

In summary, the theoretical comparison between different single-molecule tracking techniques is interesting and rigorous, but I don't think it provides enough new science to merit publication

in Nature Communications. On the other hand, the implementation of "high-density volumetric super-resolution microscopy" does not seem to contribute new science and is very poor since the systems are not optimally designed and implemented.

We thank the reviewer for their valuable comments, and we appreciate the recognition of the rigorous nature of the PSF comparisons.

The key conceptual advancement of the manuscript is 'High-density volumetric super-resolution imaging', and we hope that this is now better supported by providing

- 1) additional optical characterisation (including 7 new figures, see yellow boxes),
- 2) clear illustrations of full MLA illumination, and
- 3) edits to the main text (see red text) to better conceptually unify the manuscript.

In doing so we hope that we have remedied their concerns.

Reviewer #2 (Remarks to the Author)

The manuscript “High-density volumetric super-resolution microscopy” by Daly et al. focuses on the quantification of accessible emitter densities in single-molecule light field microscopy (SMLFM). In SMLFM, an array of micro lenses (MLA) is placed in the Fourier plane (conjugated back-focal plane) creating a pattern of PSFs that are then recorded via a camera and analysed using standard (for PSF localisation) and custom (for connecting the localisations from the different areas) software. The work is a logical extension of the author’s previous work (Sims et al., reference 22), in which the concept and the ability to image an impressive z-range of 8 μm had been demonstrated. Here, the authors used a customised MLA with a hexagonal arrangement improving the usable area on the camera. In addition to the comparison with other methodologies enabling 3D resolution in SMLM, the authors further demonstrate successful single-particle tracking over an extended biological object (B cell receptors).

The manuscript is generally well written but could, in my opinion, be considerably sharpened to provide necessary information and additional clarity to the reader. As of now, I feel that some of the comparisons with other methods are not carried out as stringent as I would have liked to see them. Overall, I still consider the manuscript a very valuable addition to the field and I hope that the authors will find the following points useful to improve the manuscript.

We thank the reviewer for their time reading the manuscript and positive response.

Major points

1. Figure 1 a/b): If I am not mistaken, the work uses a 4f arrangement of lenses as shown in one of the SI figures and not the arrangement shown here. I would suggest to merge the SI figure and the current Figure 1.

We thank the reviewer for this helpful suggestion. We have modified Fig.1 to now focus on the 3D PSFs and the light field optical implementation specifically. We have also added a new Fig. S2, which details the optical implementations of the other engineered PSFs. We hope that this now also clarifies the difference between the 4f optical configuration and the ‘3f’ optical configuration of SMLFM (where the MLA conducts the phase modification and focuses to an image plane).

2. Figure 1/2: I find it rather confusing to see different pixel sizes for different modalities in, apparently, the same field of views (in terms of constant area?). Is the reason for this a different effective magnification of the optical system? This choice makes comparing achievable emitter densities on a given camera frame area (!) rather difficult. (see also next point).

We thank the reviewer for highlighting this ambiguity and we have expanded the Supplementary Information by adding section S4.2 and Fig. S6 to provide clarification. We

hope that this makes clear that each PSF would occupy the same physical space on the detector, irrespective of pixel size, which was chosen to ensure the optimal performance of each 3D technique.

3. Figure 2a: If the FoV stays the same (10x10 μ m), the different binning would mean that even in a standard configuration 4x less molecules could be placed when a larger pixel size is used, correct?

The reviewer's feedback is appreciated here, and we would like to clarify that the simulated FoV is 20 x 20 μ m, but the figure displays a 10 x 10 μ m region for ease of visualisation. As discussed above, we have added section S4.2 and Fig. S6 to the SI to address the ambiguity around pixel size and show that density of fluorophores is limited by the size of the PSF on the detector.

4. Is Fig 2b correcting/normalising for the different magnification? It is surprising to see Light Field doing so well compared to Standard with the 4x lower number of pixels. (The focal length of both Fourier lens or MLA was given as 175mm so I would not have expected any changes in magnification, correct?)

All the comparisons are normalised for the same FoV, which is 20 x 20 μ m. Each simulation randomly positions an emitter in a 20 x 20 μ m area and implements the optimal pixel size for SNR to provide the best practical side-by-side comparison (see Table S2 in section S2.4).

5. Figure 2b: The most widely used quality of imaging metric is the Jaccard index defined as $JI = TP / TP + FP + FN$ (see, e.g. reference 6 and 12). Is there a particular reason to use precision (PPV) and sensitivity instead? If not, the authors might consider replacing sensitivity and precision with the Jaccard index to increase the overall readability of the entire manuscript.

We agree that Jaccard index is a valuable metric to include in the manuscript as it is widely used by the super-resolution community. As such, we have added Fig. S9 to the Supplementary Information, which contains plots of Jaccard index vs. density for each PSF at different photon fluences.

We chose *PPV* and *sensitivity* for the main manuscript to quantify two distinct aspects of point detection in high density settings: 1) correct identification (PPV) and 2) retrieval of points (sensitivity). Therefore, we believe that PPV and sensitivity provide more detailed information about PSF behaviour at higher densities than the Jaccard index alone.

6. Figure 2: The authors cite the work on Tetrapods by the Shechtman lab (Nehme et al, reference 6), but somewhat hide the fact that the particular publication explicitly discusses how largely overlapping PSFs can be retrieved by their deep learning

approach (DeepSTORM3D). For a truly fair comparison with the state of the art, tetrapods should be analysed with the best possible software, especially keeping in mind that the authors here, also use customised software to map the localisations. It might well be that DeepSTORM3D/DECODE on tetrapods is comparable in the achievable density without taking much away from the novelty and simplicity of the approach demonstrated here with the MLA.

We recognize the importance and complexity associated with ensuring fair treatment of each 3D PSF. The manuscript discusses the various experimental approaches to fitting at higher spot density. Here, we explore 'optical robustness', while another equally important approach is computational multi-emitter fitting (MEF)/deep learning strategies like DeepSTORM3D or DECODE. Our efforts focussed on decoupling computational MEF and optical robustness to purely quantify optical robustness. Furthermore, no computational MEF approaches currently exist for SMLFM, which prohibits fair comparison if implemented in the analysis of the other 3D PSFs. Nonetheless, the application of MEF/deep learning strategies to SMLFM would be an insightful future study.

7. Figure 5: One thing that is not yet discussed is the "real-estate" on the camera. As the camera sensor is now covering multiple field of views when using MLA, the achievable "throughput" is lower than in standard configuration. Could the authors comment on that?

The reviewer raises an interesting point because, in general, "real-estate" is not a significant issue for modern cameras, especially sCMOS. Large FoVs in STORM/PALM/PAINT require powerful lasers to encourage photo-switching, even higher powers for 3D focal volumes. 3D-SMLM is typically limited to smaller fields of view as a result, which is in good agreement with the 40 μm FoV of the platform described here (for example see <https://doi.org/10.1002/anie.202206919> or <https://doi.org/10.1038/s41467-017-02563-4>). In this work the MLA was designed to fit the whole camera chip (512 x 512 px, Evolve 512 Delta, Photometrics) but with larger, faster, sCMOS chips other additional information can be retrieved, such as splitting colour channels.

8. line 129. photon numbers: 1000 photons (and not counts, I assume) per frame for a fluorescent protein is very optimistic number to get for longer than one frame (see also next point). Also, can the authors define what they mean by "next-generation fluorescent probe"?

We used the term "next-generation fluorescent probe" to describe a conceptual fluorophore optimised for brightness that could be considered photon-unlimited. To make this clearer we have changed 'next-generation fluorescent probe' to 'photon-unlimited fluorescent probe':

A mean value of 1,000 total emitted photons per localisation for a simulated fluorescent protein was chosen because this has been reported for the likes of mEos and mMaple (<https://pubs.acs.org/doi/10.1021/acs.chemrev.7b00767>).

9. Figure 4: Tracking over such a large sample is impressive! With an average track length of 12 localisations using a very good probe, SMLFM seems rather photon hungry. So how does the approach fare with photon numbers of well below 1000 photons per localisation?

To answer this, we have added Fig. S11, which plots precision vs. photon number for this SMLFM platform. It shows that for ~800 photons per 3D localisation a precision of ~45 nm laterally and ~56 nm axially is obtained.

10. No comments were made on the availability of the customised Matlab scripts for combining the localisation data from the different FoVs. Ideally, the software will be made available on Github (or is already).

We apologise for the ambiguity here. All code and example data are available on Zenodo (<https://doi.org/10.5281/zenodo.8190164>) and a maintained version of the 3D reconstruction code on our GitHub (<https://github.com/TheLeeLab/hexSMLFM>). This includes

1. 3D reconstruction code (+ example data)
2. 3D SPT code and MLE analysis of diffusion (+ example data)
3. Matching code for simulations (+ example data)
4. Example raw imaging data (live and fixed BCR, and fixed tubulin)

To sign post this, we have added the section 'Code Availability' to the manuscript with links.

11. No comments were made on the availability of the customised MLA. Can the MLA be bought from CAIRN? Is there an estimated sales price?

MLAs, including the MLA mentioned in this document, are available from CAIRN for faster implementation. Alternatively, CAIRN can fabricate bespoke MLAs according to specifications provided by the authors, such as focal length, pitch, and number of lenses. The general fabrication cost is approximately £2000, but many companies offer commercial MLAs (we have experience also with Okotech) which can be more cost-effective.

Minor points

1. both abbreviation are present in the abstract: SMLFM and SMFLM. Please check for typos.

Amended, with thanks.

2. None of the figures of the SI were numbered in the final compiled PDF that I got. (I later saw that the SI itself shows everything correctly)

All figure numbers have been checked, with thanks.

3. Figure 1a: unit for density is missing.

We believe the reviewer is referring to Fig. 2a, which has been amended, with thanks.

4. line 125: 20x20 μm mentioned here, figure shows 10x10 μm , so 0.5 locs per 10x10 I assume?

The simulated FoV is 20 x 20 μm , the figure only shows a 10 x 10 μm FoV to enable discernment of PSF shape.

5. line 220-222: phrasing. I somewhat doubt that the division of background photons has a larger effect than adding up the camera noise contributions from adding up seven field of views. Please comment.

Thanks for the comment and to avoid confusion we have edited this sentence as follows:

“Therefore, on the basis of speed, SMLFM significantly out-performs the DHPSF at all light levels, particularly at low SNR, due to optical multi-emitter fitting.”

As background scales non-linearly with DoF in 3D-SMLM (see <https://doi.org/10.1038/nmeth.3797>) we were surprised to see such a significant speed advantage with SMLFM at low photon fluences (25.5x). Thus, in these concluding sentences of the density study, we were attempting relate the exceptional speed of SMLFM at low SNR to optical multi-emitter fitting and contrast improvement by the division of background photons. The ability to spatially localise background in the BFP to improve contrast is of functional benefit in light field and discussed in SI section S1.3.

6. I might have missed that, how was the FSC calculated (software)? 3D, as in shell, correct?

The reviewer is correct, a 3D Fourier shell correlation analysis was implemented (lines 329--331) for which we used a custom Matlab script (inspired by <https://doi.org/10.1038/nmeth.2448>).

7. line 565: What do the authors mean by creating an evanescence wave? If I am not mistaken, TIRF would require an oil immersion objective with an NA of >1.4.

We have removed ‘evanescent wave’.

8. I am surprised to see the mentioning of #1 cover slips. In most cases, the objectives are designed for #1.5 to minimise aberrations etc. Worth checking.

We have amended as follows:

“Glass slides (VWR, 631–1570) were washed with propan-2-ol and water...”

The coverslips used are VWR 631-1570, No. #1 with a thickness of 0.13–0.16 mm. Aberrations were minimised by careful adjustment of the objective correction collar.

9. line 575 unit missing.

Amended.

10. how “easy” is it to adapt the software/noise model for sCMOS rather than emCCD?

This would be very easy with the only key difference being per-pixel gain. As SMLFM uses existing localisation algorithms this is easily accounted for and compatible with all code implemented in this study.

Reviewer #3 (Remarks to the Author)

Daly et al present a new approach to volumetric SMLM imaging, splitting the light according to its position in the back focal plane and thereby forming multiple images, each taken at a different angle. This is a potentially very useful new approach; while many solutions have been suggested to the challenge of how to discriminate axial position, astigmatism is still the most commonly used method despite its limitations (limited z depth in a single slice, degradation of x or y precision). In particular, the authors demonstrate that their method allows use of substantially higher excitation densities than can typically be used in 3D imaging. Overall this paper represents an important and interesting new technique that could help uncover exciting new biology. However, I have some comments that I would like to be addressed.

We thank the reviewer for their positive comments and recognizing SMLFM as an important technique for uncovering new biology.

While the acquired data is impressive the imaging of B cell receptors have a key limitation. Given the size of the cell and the number of molecules localised, I estimate that if the molecules are evenly spaced they are around 70 nm apart. They are of course not evenly distributed, but it illustrates that the resolution is likely to be limited in at least some areas by the sampling rather than localisation precision (a common problem when imaging with dSTORM in thick samples). The acquired density of molecules can be limited if only a small amount of the labelled molecule is present, or by the labelling efficiency, the dSTORM blinking degrading, or the acquisition time: the authors should comment on what limited the reconstructed density in this case.

The reviewer's comments regarding sampling rate in the BCR images is greatly appreciated. To clarify, all dSTORM imaging of B cells was conducted until all blinking events were exhausted and therefore the accumulation of localisations was limited by the abundance of receptors. It is for this reason that we also image highly dense tubulin datasets in HeLa cells. Here the labelling density is significantly greater, and hence these datasets were used for resolution analysis.

While the BCR imaging might not lead to as high spatial sampling as the tubulin imaging, the rate of localisations over a small detector area illustrates the power of SMLFM at resolving single emitters at high density though optical multi-emitter fitting. Therefore, we believe that combining these two experiments with live-cell tracking covers most scenarios that necessitate high-density volumetric SMLM.

I generally think it is important to benchmark against a known structure in an experiment when presenting new techniques. However in this case, as the focus is on accurate recall at high density rather than resolution per se, I think that the evaluation is satisfactory without additional experiments.

We thank the reviewer for the suggestion and hope that the study into the PPV and sensitivity of SMLFM in combination with the imaging of well-known tubulin architectures, provides confidence in the resolving power of SMLFM.

Minor concerns

1. With regard to the double helix point spread function, a brief mention of the tradeoff between PSF size and DoF for this particular technique would be helpful.

We have modified the manuscript as follows:

“Compared to the other techniques, SMLFM differs in that it breaks the observed trade-off trend between PSF size and DoF. This allows for an axial range that is suitable for imaging entire cells up to 8 μm , with a PSF area that is on average 55% the size of the DHPSF.”

Furthermore, we have expanded Fig. S5 to accompany this discussion, which now includes a plot of PSF footprint vs. DoF. In the plot we further confirm the trend-breaking nature of SMLFM. The 3D PSFs (including the DHPSF) fall along a general positive (non-linear) curve, except for SMLFM, which has a substantially larger DoF than expected.

2. “this work will focus on techniques that yield sub-diffraction axial precision over extended axial ranges” – it needs to be clarified that the authors mean single shot techniques, stitching together multiple reconstructed planes is a standard approach to this problem.

We have edited the manuscript as follows:

“However, this work will focus on techniques that yield single-snapshot sub-diffraction axial precision over extended axial ranges.”

3. In Figure 3a the standard and astigmatic curves look flat but I think would in fact vary. Could these curves also be shown with the vertical axis expanded so this can be seen?

We believe the reviewer is referring to Fig. S3a and thank them for this suggestion. We have expanded the astigmatic plot, now in Fig. S5b, to show the astigmatic curve more clearly.

4. In Figure 2 I found the labels giving the DoF in panel B confusing, I think the figure would be clearer with them removed.

We have moved the DoF labels to the left-hand side of the figure.

5. In Figure S5 the dark grey and black lines cannot be clearly distinguished, the lines should either be colour or separated into individual graphs.

We have now coloured the plot differently and increased opacity to make the individual trends clearer.

6. I don't think the PPV/sensitivity values are particularly helpful for the experimental data since this is from simulation. The expected high performance of the technique has been made clear in the part of the paper covering the simulations.

We have amended the manuscript by removing the discussion of PPV and sensitivity from the practical half of the text.

7. The FSC measurement seems lower than I would expect (while the figure may seem quite high for SMLM accurate FSC measurements tend to come out substantially higher than other resolution metrics), probably due to the calculation not taking into account multiple localisations of the same fluorophore. This is a known challenge in dSTORM which can bias measurements, and if it has not been corrected for then a note (e.g. no correction for fluorophore reappearance) should clarify that.

The reviewer is correct that the image resolution and localisation precision are convolved without temporal grouping. We thank them for pointing this out and have temporally grouped all dSTORM data to deconvolve these effects. This gives a new value of 59 nm for the FSC, shown in Fig. S17. We have updated the Methods section and added a new Fig. S13 to describe the temporal grouping pipeline. We have also amended the manuscript text to clarify the use of temporal grouping and reproduced the data in Fig. 3 (BCR dSTORM data).

8. In the image of the microtubules the upper part of the super-resolution image appears substantially degraded compared to the lower part. Why is this? Was it due to issues keeping the excitation density low in this area? Or was this at the edge of the imaging area and this caused a degradation in quality?

We have re-processed and modified Fig. 5 in response to this. The resolution is linked to the trade-off between photoactivation intensity and the time scale of the experiment, and the close proximity to the imaging area to the nucleus of the cell. The objective of the figure is to provide evidence that the microtubules are readily super-resolved at high labelling densities (up to 40 locs per frame) in a well-known biological structure.

REVIEWERS' COMMENTS

Reviewer #2 (Remarks to the Author):

The authors have addressed all points raised in my previous assessment. I consider the manuscript now suitable for publication.

Reviewer #3 (Remarks to the Author):

The authors have amended the paper in response to my comments and I now consider the paper suitable for publication in Nature Communications.

Reviewer #4 (Remarks to the Author):

In this manuscript, the authors demonstrate high-density volumetric super-resolution microscopy using a technique termed single-molecule light field microscopy (SMLFM). SMLFM is a 3D single molecule localization microscopy which has already been reported in the author's previous Optica paper. In the prior work, they have achieved up to 20 nm localization precision throughout an extended 6 μ m DOF. In current work, the authors report the implementation of single-snapshot 3D super-resolution imaging over an 8 μ m DoF. So the difference lies in the single-snapshot implementation using a hexagonal MLA. That might be the reason why the Reviewer 1 didn't see anything new. The authors claim their key conceptual advance in this work was a high density 3D super-resolution technique. I don't think so because their prior work could also do it. However, it should be noted that they first demonstrated the ability of high-density 3D super-resolution microscopy in this work. Overall, I still consider the manuscript a useful addition to the field.

The author present two evidences to demonstrate high-density volumetric super-resolution imaging: computational comparison between different single-molecule tracking techniques and experimental demonstration of single-particle tracking over an extended biological object. The results are convincing and look good. But if the author can provide an experimental evidence that SMLFM outperforms the commonly-used astigmatism method, it will be perfect.

A technical point: I also find the strong overlap between the views. The authors take it as non-uniform background signal, but I don't think so. It would be better to figure out the real reason.

Another point about the abstract. When I first read the abstract, I think this work improves the temporal resolution and the authors claimed they achieved an order-of-magnitude speed improvement compared to the double helix PSF. Finally, I find that it is not the temporal resolution but the density. Although they are related, they are actually totally different. Mostly importantly, the authors didn't demonstrate the improvement in temporal resolution. So it would be better to rewrite the abstract part and clearly present the real achievement in this work.

RESPONSE TO REVIEWERS' COMMENTS

General remarks to reviewers are in blue; changes to the manuscript are in italicised blue.

Reviewer #2 (Remarks to the Author):

The authors have addressed all points raised in my previous assessment. I consider the manuscript now suitable for publication.

Reviewer #3 (Remarks to the Author):

The authors have amended the paper in response to my comments and I now consider the paper suitable for publication in Nature Communications.

Reviewer #4 (Remarks to the Author):

In this manuscript, the authors demonstrate high-density volumetric super-resolution microscopy using a technique termed single-molecule light field microscopy (SMLFM). SMLFM is a 3D single molecule localization microscopy which has already been reported in the author's previous *Optica* paper. In the prior work, they have achieved up to 20 nm localization precision throughout an extended 6 μm DOF. In current work, the authors report the implementation of single-snapshot 3D super-resolution imaging over an 8 μm DoF. So the difference lies in the single-snapshot implementation using a hexagonal MLA. That might be the reason why the Reviewer 1 didn't see anything new. The authors claim their key conceptual advance in this work was a high density 3D super-resolution technique. I don't think so because their prior work could also do it. However, it should be noted that they first demonstrated the ability of high-density 3D super-resolution microscopy in this work. Overall, I still consider the manuscript a useful addition to the field.

We thank the reviewer for their time spent reading the manuscript and for considering it a useful addition to the field.

The author present two evidences to demonstrate high-density volumetric super-resolution imaging: computational comparison between different single-molecule tracking techniques and experimental demonstration of single-particle tracking over an extended biological object. The results are convincing and look good. But if the author can provide an experimental evidence that SMLFM outperforms the commonly-used astigmatism method, it will be perfect.

This is a very helpful point, which we think is a result of a conceptual confusion, which prohibits the direct experimental comparison between astigmatism and SMLFM that the reviewer is asking for. We have changed the manuscript text to clarify this (see below).

In brief, the technical differences can be summarized by two key points (1) Effective point picking and (2) Technical implementation. Astigmatism and SMLFM perform similarly at (1), but the difference in (2) prohibits a fair and direct comparison.

1. **Point picking.** Our simulations—in which the number of emitters is controlled to generate localisation data—indicate that astigmatism and SMLFM are similarly effective at identifying peaks and filtering noise, as shown in Fig. 2 and Fig. S11 at an emitter density of 0.1 locs μm^{-2} (c.f. BCR imaging density).
2. **Technical implementation.** The primary difference between astigmatism and SMLFM lies in their axial ranges, approximately 800 nm in practice for astigmatism¹⁻³ and 8 μm

for SMLFM, a 10× increase. Additionally, SMLFM benefits from an optical redundancy by capturing 3D volumes through multiple perspective views, making it particularly suitable for dense single-molecule datasets.

Consequentially, while both techniques offer comparable resolving power, SMLFM enhances throughput in the imaging of whole cells. It achieves this by eliminating the need for axial ‘plane’ scanning, therefore reducing out-of-plane photobleaching, and removing the necessity for different labelling methods/complex buffer replenishment, etc.⁴ Given these differences, a direct experimental comparison of the two methods is not quantitatively feasible, other than our in-depth simulated comparison described in Fig. 2 over each respective imaging volume. The novelty of our paper lies in the combination of these factors, which makes high cellular throughput (n = 10 to 100) now practical with SMLFM

We have changed the text to make this more explicit:

Lines 251—261: “Our simulations indicate that at a localization density of $0.10 \mu\text{m}^{-2}$, both SMLFM and astigmatism (most commonly used) achieve equal sensitivity, $86.6 \pm 0.9\%$ and $86.4 \pm 0.5\%$ (see Fig 2b), and a Jaccard index of $86.0 \pm 0.9\%$ and $82.6 \pm 0.6\%$, respectively (see Supplementary Fig. S11). The comparable resolving power of both techniques combined with the 8-fold larger depth-of-field afforded to SMLFM, eliminates the need for axial scanning, and elevates SMLFM to a region of high biological throughput and applicability. Importantly, SMLFM is advantageous at *these* high densities because single emitters are not required to be isolated in every perspective view to be localized in 3D.”

A technical point: I also find the strong overlap between the views. The authors take it as non-uniform background signal, but I don’t think so. It would be better to figure out the real reason.

We thank the reviewer for highlighting this point for further clarity, which we responded to by updating the example data during the 1st revision. We have now also produced new Supplementary Figure S4 to clarify the effect of non-uniform background. (reproduced below)

The new example data (.tif stacks) shows raw tubulin imaging data with a mechanical iris mounted in the image plane and the previous apparent overlap no longer present. Supplementary Figure S4 illustrates the combined effect of an iris placed at the image plane under epifluorescence (Epi) and an inclined illumination (HILO) to show non-uniform

background (scale bar 20 μm). In cellular systems the background will exhibit further inhomogeneity as originally pointed out by Reviewer 1.

Another point about the abstract. When I first read the abstract, I think this work improves the temporal resolution and the authors claimed they achieved an order-of-magnitude speed improvement compared to the double helix PSF. Finally, I find that it is not the temporal resolution but the density. Although they are related, they are actually totally different. Mostly importantly, the authors didn't demonstrate the improvement in temporal resolution. So it would be better to rewrite the abstract part and clearly present the real achievement in this work.

We thank the reviewer for pointing this out, and we agree that “temporal resolution” could be misinterpreted here. We have therefore made the following edits to the abstract and introduction to eliminate this ambiguity:

- Lines 4—7: “However, the resulting large and complex PSF spatial footprints reduce biological *throughput and applicability* by requiring lower labelling densities to avoid overlapping fluorescent signals”
- Lines 40—42: “However, the number of emitters localised per frame governs *imaging speed* and therefore dense emitter datasets are desirable”

We used “temporal resolution” to describe an increased number of fluorophores localised per unit time (“the number of emitters localised per frame governs temporal resolution and therefore dense emitter datasets are desirable”, line 40) and hence have edited this.

We thank the editor and reviewers for their guidance improving the manuscript.

References

- 1 W. R. Legant, L. Shao, J. B. Grimm, T. A. Brown, D. E. Milkie, B. B. Avants, L. D. Lavis and E. Betzig, *Nat. Methods*, 2016, **13**, 359–365.
- 2 A.-K. Gustavsson, P. N. Petrov, M. Y. Lee, Y. Shechtman and W. E. Moerner, *Nat. Commun.*, 2018, **9**, 123.
- 3 C.-H. Lu, W.-C. Tang, Y.-T. Liu, S.-W. Chang, F. C. M. Wu, C.-Y. Chen, Y.-C. Tsai, S.-M. Yang, C.-W. Kuo, Y. Okada, Y.-K. Hwu, P. Chen and B.-C. Chen, *Commun. Biol.*, 2019, **2**, 1–10.
- 4 N. E. Albrecht, D. Jiang, V. Akhanov, R. Hobson, C. M. Speer, M. A. Robichaux and M. A. Samuel, *Cell Rep. Methods*, 2022, **2**, 100253.

REVIEWERS' COMMENTS

Reviewer #4 (Remarks to the Author):

The authors have addressed my concerns, and thus I consider the manuscript now suitable for publication.